# FedHC: Proximal Correction with Hessian and Cosine Correlation for Federated Learning paper Id: 3657

## Abstract

Federated learning (FL), a prominent distributed learning approach, involves collaborative updates among participants and individual updates on private data. While widely-used FL methods, such as FedDC and others, traditionally rely on first-order optimization techniques like Stochastic Gradient Descent (SGD) to achieve convergence, there is a growing interest in leveraging second-order optimization methods to enhance convergence in complex models. However, applying these second-order techniques to FL models often results in convergence challenges. To address these issues, we present an innovative integrated methodology known as FedHC, combining proximal correction with Hessian optimization and cosine correlation for FL. FedHC introduces the Hessian optimizer with proximal correction to accelerate convergence. Additionally, we employ cosine correlation to minimize learning discrepancies and bridge the gap between local and global models. Experimental results and analyses conducted on four datasets demonstrate that FedHC significantly accelerates convergence and outperforms existing methods in various image classification tasks, maintaining robustness in both IID and Non-IID client settings.

## 1 Introduction

Deep learning algorithms have made impressive progress owing to the abundance of large-scale data Lin et al. (2014); Cordts et al. (2016). However, in real-world scenarios, data is often distributed among different clients, including mobile devices and organizations. Due to increasing concerns about privacy and strict data protection regulations Voigt & Von dem Bussche (2017), these clients can not pool their data to train a model collectively. Driven by such realistic issues, Federated Learning (FL) approaches Yang et al. (2019); McMahan et al. (2017a) have emerged as a privacy-preserving approach.

FL enables multiple participants to cooperate in training a global model collaboratively in a decentralized manner without leaking private data. This novel approach represents a significant advancement in deep learning paradigms, offering new possibilities for secure and collaborative model training across diverse data sources. FL has become a vibrant and challenging research area, showcasing promising results in real-world applications. Albeit immense progress, researchers have encountered noteworthy challenges Kairouz et al. (2021); Li et al. (2020a) in FL. A prominent and inevitable challenge is the heterogeneity problem arising from both statistical and systematic differences among the participating clients. Statistical heterogeneity encounters due to non-independent and identically distributed (Non-IID) data. *This Non-IID data distribution leads to inconsistencies in clients' local objective functions and optimization directions, making achieving efficient and accurate model training more complex.* The distinct data distributions create a fundamental discrepancy in achieving minimum local empirical loss while simultaneously reducing the global empirical loss. This discrepancy poses a challenge in highly heterogeneous environments, where FedAvg McMahan et al. (2017b) lacks convergence assurance. It only attains compromised rates of convergence speed and model performance. Research studies Karimireddy et al. (2020); Khaled et al. (2020) have proven that data heterogeneity causes drift in clients' local updates, consequently leading to a deceleration in convergence speed. The drift in parameters between a client's model and the centralized learning model arises as a result of two factors: residual parameter drift carried over from the previous round and the gradient drift occurring in the current round Zhao et al. (2018).

To resolve the issue of client drift in FL, researchers have come up with different approaches Karimireddy et al. (2020); Li et al. (2019). Li et al. (2020c) proposed a FedProx approach, where a proximal term is utilized to reduce the differences between the local and global models. This makes the local updates less variable and brings them closer to the global model. However, there's a trade-off with this approach, as the introduced proximal term helps reduce drift, but it may hinder the global model from reaching its optimal global stationary point. To overcome the problem of client drift, Karimireddy et al. (2020) provided a solution in Scaffold. The Scaffold used a control gradient variate to reduce the drifting of gradients in each communication round. Gao et al. (2022) used gradient correction term with variance reduction to dynamically update the loss function for overcoming the drift. However, it is unable to completely eliminate the drifting and some residual deviation between local and global models. Consequently, residual deviations accumulate during the training process, causing slower learning leading Zhao et al. (2018).

In many of the earlier FL approaches, the focus was on ensuring that the local models were consistent with the global model to reduce gradient drift. These FL approaches were successful to some extent in reducing the drift but enlarged the deviation between local and global model parameters. To address this problem, Gao et al. (2022) decoupled the local and global models by establishing a relationship among them using constraint penalty terms. Nevertheless, the disparity between the local and global models has not been entirely resolved. Considering the fundamental discrepancy between local and global optimal points in FL, we propose a novel proximal correction with Hessian and cosine correlation in federated learning (FedHC). In FedHC, we introduce a cosine similarity correlation to establish a harmonious relationship between global parameters and local parameters to reduce the disparities between global and client models.

Deep learning techniques frequently leverage similarity measures such as cosine similarity to boost model generalization. Cosine similarity, which primarily focuses on vector directions, is notably effective in Natural Language Processing (NLP) tasks, especially when confronted with vastly different word frequency magnitudes. We noticed considerable gradient inconsistencies in the Scenario among clients. To rectify this, we implemented cosine similarity correlation, which has proven efficient through mathematical proofs and experimental results. In addition, we propose an approximated Hessian optimizer to elevate the training with fast convergence. By the synergistic effect of both cosine correlation and approximated Hessian optimizer, the proposed FedHC effectively reduces the drift between local and global models, achieving faster convergence compared to state-of-the-art (SOTA) FL algorithms. Our main contributions are as follows:

1. An approximated Hessian optimizer is proposed to elevate and optimize the training models with fast convergence for federated learning.
2. A proximal cosine correlation is introduced in the objective function to mitigating the disparities between the global and client models.
3. We also integrated the Hessian diagonal operator and cosine correlation to strengthen the connection between global and local models and to promote collaborative learning for fast convergence.

We have verified empirical convergence results on various public datasets, such as MNIST, EMNIST-L, CIFAR10, and CIFAR100 datasets. The results demonstrate that the proposed FedHC outperforms SOTA FL methods (*e.g.*, FedDC Gao et al. (2022); FedDyn Acar et al. (2021); Scaffold Karimireddy et al. (2020); FedProx Li et al. (2020b); and FedAvg McMahan et al. (2017b)) in both IID and Non-IID client settings, achieving the best performance with significantly faster convergence.

## 2 RELATED WORK

Federated Learning (FL) is a rapidly evolving topic involving mainly two types of updates: server and device. In FL, the goal is to minimize a local loss function associated with each update, which can be dynamically updated over different rounds. Some methods aim to fully optimize the updates, while several methods propose inexact optimization Konečný et al. (2016); Kairouz et al. (2021).

FedAvg McMahan et al. (2017b) is a pioneering work that uses weighted parameter averaging to update parameters from multiple clients. It has been shown in works Patel & Dieuleveut (2019); Khaled et al. (2020) that FedAvg achieves asymptotic convergence in scenarios involving homogeneous clients. However, Woodworth et al. (2018) analyzed that the convergence bounds of FedAVG

can exhibit notable variations for heterogeneous clients. Researchers Li et al. (2019); Karimireddy et al. (2020) suggest that client drift caused by Non-IID data is the prime factor affecting heterogeneous client's convergence speed. Previous research Li et al. (2019); Karimireddy et al. (2020) highlighted the challenges introduced by Non-IID data, such as gradient divergence, biases in optimization, and unassured convergence. Some approaches Kim et al. (2022); Wang et al. (2023) aim to reduce the variance of clients' updates to accelerate convergence. The empirical risk function may be minimized using a uniform global model across clients with non-IID distributed data, however, making convergence to an optimal global model becomes a more difficult task Li et al. (2020b). FedProx Li et al. (2020b) addresses statistical heterogeneity and improves stability by introducing proximal regularization to the local model with the global model. This proximal term ensures that the updated local parameter remains close to the global model, thereby mitigating the risk of gradient divergence. Nevertheless, this approach overlooks the distinction between the optimal points of local empirical objectives and the global optimal point, which can result in suboptimal performance. An important drawback of these methods is their failure to account for differences in client models, leading to suboptimal performance and slower convergence rates, especially in scenarios involving Non-IID data distributions.

Scaffold Karimireddy et al. (2020) modified the gradients for each client to overcome the client drift between local and global models. Likewise, FedDyn Acar et al. (2021) urges an adaptive regularizer for each client to align the global and client model parameters and reduce communication overhead. Another line of research focuses on optimizing the server's parameter in the aggregation step to obtain a significantly better model. Work Zhang et al. (2020) dynamically computes the optimal weighted clients model for the creation of a global model. Reddi et al. (2020) federated adaptive optimization technique that considers client heterogeneity and communication efficiency to prevent undesirable convergence behaviour. It builds upon the work of Yang et al. (2021), which achieved linear speedup with Non-IID data by using two-sided learning rates in local and global updates. These approaches can be seamlessly integrated into existing methods and have demonstrated improved convergence speed and performance compared to FedAvg. However, Zhao et al. (2018) presented a theoretical concern that parameter deviation accumulates and leads to suboptimal solutions in federated learning settings. Gao et al. (2022) strives to minimize the variances in global, local model parameters, and previous local gradients and expected gradient values. To address these issues, the proposed FedHC bridges the gap between local and global models by establishing a cosine correlation. Also, the proposed FedHC uses proximal correction with an approximated Hessian optimizer to ensure better convergence.

## 3 FEDHC- FEDERATED OPTIMIZATION USING HESSIAN AND COSINE CORRELATION

The SOTA FL methods, FedAvg Mills et al. (2019); FedProx Li et al. (2020c); Scaffold Karimireddy et al. (2020); FedDyn Acar et al. (2021); and FedDC Gao et al. (2022); employ first-order optimization with SGD to achieve convergence. However, research indicates 202 (2020) that second-order optimization techniques, such as Hessian optimization, can enhance the convergence of complex models. Nonetheless, employing second-order optimization with the current FL (FedDC, FedDyn, *etc.*) model results in non-convergence (as evident from experiments, Section 4) due to the nature of existing objective functions. This issue arises from the linear nature of the gradient correction and dynamic regularizer components within the objective function of FedDC and FedDyn, causing them to lose significance during the second-order optimization process. This demands necessary adjustments to the objective function to effectively harness the capabilities of second-order optimizers. To address this and inspired by the successful application of cosine correlation in Natural Language Processing (NLP) tasks, we propose a unique integration method that combines the Hessian operator with cosine correction. This innovative approach has proven successful in achieving model convergence within the context of FL. Furthermore, we consider utilizing an approximated Hessian optimizer to balance computational complexity without significantly compromising performance. The primary objective of this paper is to minimize the global loss by optimizing the global model parameters across $D$ datasets from $N$ clients. To achieve this, we employ a second-order optimizer on the local parameters as follows.

$$\text{argmin}_{\theta \in \mathbb{R}^N} \left\{ L^t(\theta) = L_i^t(\theta_i) : \theta = (\theta_i) \text{ is the local model parameters at round } t. \right\} \tag{1}$$

**Definition 3.1** *A point $\theta^* = (\theta_i^*) \in \mathbb{R}^N$ is said to be a argmin point of $L^t$ if*

$$L_i^t(\theta_i^*) \leq L_i^t(\theta_i), \quad \forall \theta = (\theta_i) \in \mathbb{R}^N, \text{ and at any time } t.$$

The necessary condition for optimal points is the following:

**Theorem 3.1** *Assume that the loss function $L$ is a continuous twice-differentiable function. Then $L$ attains a minimum value at point $\theta^*$ if*

$$\nabla(L_i(\theta_i^*)) = 0 \ \forall i \in N \text{ and } |H(L_i(\theta_i^*))| > 0,$$

*here, $|H|$ is the determinant of the Hessian operator.*

More details about the definitions, theorems, and equations are presented in the supplementary document.

## 3.1 Objective function using the Cosine correlation

In FL, a federation is formed by N clients, each having its private local dataset denoted as $D_i$. We select $C_t, (C_t \subseteq [N])$ active clients, and share the global model $w$ for training using client data. Then, each client computes the loss and updates the model parameters using four components: an empirical loss, a drift-based penalty, a proximal gradient correlation, and a Hessian optimizer. The drift-based penalty and the proximal gradient correlation are employed to enhance the optimization capability of the loss function. This optimization addresses the intricacies of data heterogeneity among clients, amplifying the learning process's overall efficiency and effectiveness.

Specifically, each client computes the drift value $d_i$, under the constraint that satisfies the condition: $d_i = w - \theta_i$, where $\theta_i$ represents the client $i^{th}$ local model. Maintaining this constraint throughout training is paramount to effectively controlling the local drift variable. To achieve this, we transform the constraint into a penalized term as follows $P_i$:

$$P_i(\theta_i, d_i, w) = \|d_i + \theta_i - w\|^2, \quad \forall i \in [N] \tag{2}$$

Here, $\|\cdot\|$ represents the $\ell_2-$norm of a vector and $[N]$ denotes the set of all clients.

Each client can effectively update its model parameters and local drift variables by integrating this penalized term with the empirical loss term on their respective client's dataset. As a result, we convert an equation-constrained optimization problem into an unconstrained optimization problem. In the proposed FedHC approach, the objective function consists of three main components: the local empirical loss term $\mathcal{L}(\theta_i)$, the penalized term $P_i$, and proximal gradient correction $S_i$. Specifically, for client $i$ (where $i \in [N]$), the local objective function is as follows:

$$L_i^t(\theta_i, d_i, D_i, w) = \mathcal{L}(\theta_i) \ + \ \frac{\alpha}{2} \ P_i(\theta_i; d_i, w) \ + \ S_i(w, \theta_i, g_i, g) \tag{3}$$

where, $L_i^t(\theta_i, d_i, D_i, w)$, is objective function to optimize $i^{th}$ client in $t^{th}$ round, $\mathcal{L}(\theta_i)$ represents the ordinary empirical loss, The $P_i$ is the penalized term as defined in Eq. 2, and $(\alpha)$ is a hyperparameter that controls the weight of the penalized term.

Furthermore, drawing inspiration from Scaffold Karimireddy et al. (2020), we introduce a proximal gradient correction, $S_i$, to support the second-order optimization. The proximal cosine correction involves the $(\cdot)$ product of the cosine correlation and the variance of the last local model parameters. As a result, the cosine correlation effectively identifies the differences between the current local and global models. The detailed definition of cosine correlation is present in *Definition* 3.2.

**Definition 3.2** *The cosine correlation between local $(\theta_i)$ and global $(w_i)$ parameters for the $i^{th}$ client $S_i : \mathbb{R}^4 \to \mathbb{R}$ is defined as*

$$S_i(w_i, \theta_i, g_i, g) = \frac{1}{\eta K} \left( \left( 1 - \frac{w \cdot \theta_i}{\|w\| \cdot \|\theta_i\|} \right)^2 (g - g_i) \right) \tag{4}$$

*Here, $\eta$ is the learning rate, $K$ is the number of training iterations in one round, $g_i$ denotes the local update value of client $i$'s local parameters in the last round, while $g$ represents the estimated update value of all the clients' local parameters in the previous round. In the $t^{th}$ round, we have $g_i = \theta_i^t - \theta_i^{t-1}$ and $g = \mathbb{E}_{i \in N}(g_i)$, where $\theta_i^t$ and $\theta_i^{t-1}$ are client i's local model parameters in the $t^{th}$ round and $(t-1)^{th}$ round, respectively.*

The term $S_i$ aims to reduce the disparities at local and global gradients, thereby contributing to a more stable and efficient training process. The detailed structure for FedHC is outlined in Algorithm 3.2.

The following Lemma proves the smoothness of the cosine correlation term $\frac{1}{2^k}$, where $k$ is the number of epochs on the client side.

**Lemma 3.1** $\|S_i(w_1, \theta_i, g_i, g) - S_i(w_2, \theta_i, g_i, g)\| \leq \frac{1}{2^k}\|w_1 - w_2\|^2 \; \forall \theta_i, g_i \text{ and } g, \; i \in [N]$.

### 3.2 THE LOCAL MODEL PARAMETERS UPDATE USING APPROXIMATED HESSIAN OPTIMIZER

In FL, the global model $(w)$ is initially distributed to all clients $i$ ($\forall i \in [N]$). Subsequently, each client proceeds to train this global model using its respective local dataset by minimizing the objective function outlined in Eq 3. The gradients of the objective function $\gamma_i^{t,k}$ in the $t^{th}$ global round and $k^{th}$ local iteration are calculated as follows.

$$\gamma_i^{t,k} = \frac{\partial L(\theta_i^{t,k}, d_i, D_i, w^t)}{\partial \theta_i^{t,k}} \tag{5}$$

By assuming the Hessian of the objective function is positive-definite, we get the descent direction can be represented as a positive-definite Hessian $H_i^{t,k}$.

**Definition 3.3** *The Hessian optimizer for federated learning is defined as:*

$$\nabla \theta_i^{t,k} = H_i^{t,k,-p} \gamma_i^{t,k} \tag{6}$$

*where the Hessian decomposition is given as:* $H_i^{t,k,-p} = U_i^T \Lambda_i^{-p} U_i^{t,k}$.

The value $p$ ($0 < p < 1$) is known as the "Hessian power". The matrix $U_i^T \Lambda_i U_i^{t,k}$ presents the eigen decomposition of the Hessian matrix $H_i^{t,k}$. It's important to note that when the value of $p$ is set to 1, the optimizer behaves similarly to the Newton method, while for $p = 0$, it behaves more like the regular gradient descent method.

The novel idea in the second-order Hessian-based optimizer for the loss function involves reloading the gradient $\gamma_i^{t,k}$ onto the Hessian $H_i^{t,k,-p}$. It is well known that determination of the Hessian is a daunting task and is computationally expensive. To reduce the computational cost associated with the second-order optimization, we replaced the Hessian $H_i^{t,k,-p}$ with its diagonal counterpart. This transformative tweak unveils the following equation:

$$\nabla \theta_i^{t,k} = Diag(H_i^{t,k,-p}) \, \gamma_i^{t,k} \tag{7}$$

To compute this diagonal, we employ Hutchinson's method Yao et al. (2018).

It is widely acknowledged that Hutchinson's method efficiently computes the Hessian's diagonal matrix by utilizing the Hessian-free method as a resource for Hessian vector products, all at a reasonable computational expense.

In particular, the Hessian-free method acts as a medium between the Hessian matrix with a random vector $z$ via chain rule as follows:

$$\frac{\partial \gamma_i^{t,k} z}{\partial \theta_i} = \frac{\partial \gamma_i^{t,k}}{\partial \theta_i} z + \gamma_i^{t,k} \frac{\partial z}{\partial \theta_i} = \frac{\partial \gamma_i^{t,k}}{\partial \theta_i} z = \mathbf{H_i^{t,k,-p}} z \tag{8}$$

Here, $z$ represents a random vector (independent of $\theta_i$) following the Rademacher distribution. This efficiently computes $\mathbf{H_i^{t,k,-p}} z$ without having the explicit form of the Hessian, by back-propagating the $\left(\gamma_i^{t,k}\right)^T z$ term. This reduces the computational cost Yao et al. (2018). Further, the Hessian diagonal is obtained by using the following Hutchinson's method:

$$Diag(H_i^{t,k,-p}) = \mathbb{E}[z \odot (H_i^{t,k,-p} z)] \tag{9}$$

---

**Algorithm 1** `Proximal correction with Hessian and cosine correlation for federated learning`

---

**Input:** Randomly initialize the global model- $w$, learning rate- $\eta$, training global rounds- T, local iterations- K, initially local drift $h_i = 0$, $N$- total number of clients, $C_t$- fraction of selected clients

---

**Output:** $w^*$ minima of global empirical loss

---

1: **for** $t = 1, 2, 3...T$ **do**
2:     Sample the active client set $C_t \subseteq [N]$
3:     **for** client $i \in C_t$ in parallel **do**
4:         Set the local model parameter $\theta_i := w$
5:         **for** $k = 1, 2, ..., K$ **do**
6:             Compute the local loss $L_i^{t,k}(\theta_i, d_i, D_i, w)$
7:             Compute the local gradient $\gamma_i^{t,k} = \eta \frac{\partial L_i^{t,k}(\theta_i; d_i, D_i, w)}{\partial \theta_i}$
8:             Calculate the first and second-order momentum $m_i^{t,k}$ and $v_i^{t,k}$
9:             $m_i^{t,k} = \frac{(1-\beta_1)\sum_{k=1}^{K}\beta_1^{K-k}\gamma_i^{t,k}}{1-\beta_1^k}$
10:           $v_i^{t,k} = \sqrt{\frac{(1-\beta_2)\sum_{k=1}^{K}\beta_2^{K-k}Diag\left(H_i^{t,k,-p}\right)^s Diag\left(H_i^{t,k,-p}\right)^s}{1-\beta_2}}$
11:           Update the local model parameters: $\theta_i^{t,k+1} = \theta_i^{t,k} - \eta \frac{m_i^{t,k}}{v_i^{t,k}}$
12:         **end for**
13:         Set the local gradient drift $\Delta\theta_i = \theta_i - w$
14:         Update the local gradient drift: $d_i = d_i + \Delta\theta_i$
15:     **end for**
16:     Update the global model: $w = \frac{1}{|C_t|}\sum_{i \in C_t}(\theta_i + d_i)$
17:     Set global gradient drift $\Delta\theta = \frac{1}{|C_t|}\sum_{i \in C_t}\Delta\theta_i$
18: **end for**

---

Hutchinson's method enables the computation of the Hessian diagonal by using the expectation of z $\odot(H_i^{t,k,-p}z)$ Bekas et al. (2007). Here $\odot$ represents component-wise multiplication of vectors.

The Hessian diagonal might vary significantly across each unique parameter dimension in the underlying problem. To diminish this, we perform spatial averaging $\left(Diag(H_i^{t,k,-p})\right)^s$ of the Hessian diagonal. Further, the responses of smooth spatial averaging will be used in the Hessian diagonal momentum. This second-order moments, denoted as $v_i^{t,k}$ and computed as:

$$v_i^{t,k} = \sqrt{\frac{(1-\beta_2)\sum_{k=1}^{K}\beta_2^{K-k}\left(Diag(H_i^{t,k,-p})\right)^s \left(Diag(H_i^{t,k,-p})\right)^s}{1-\beta_2}} \tag{10}$$

Instead of applying only a second-order momentum, the Hessian diagonal momentum, FedHC uses a first-order gradient momentum $\left(m_i^{t,k}\right)$ and Hessian diagonal momentum $\left(v_i^{t,k}\right)$ to smooth out local variations in the gradient update. The first-order momentum is computed as follows:

$$m_i^{t,k} = \frac{(1-\beta_1)\sum_{k=1}^{K}\beta_1^{K-k}\gamma_i^{t,k}}{1-\beta_1^k} \tag{11}$$

In continuous with the previous discussion, the proposed FedHC model parameters are updated as follows:

$$\theta_i^{t,k+1} = \theta_i^{t,k} - \eta \frac{m_i^{t,k}}{v_i^{t,k}} \tag{12}$$

### 3.3 UPDATING THE LOCAL DRIFT AND GLOBAL MODEL PARAMETER

The proposed FedHC method uses the local drift variables, which are used to quantify the deviation of each client's local model from the global model. These drift variables are computed by taking the

difference between the local and global model parameters, and are used to adjust the local model parameters before sending them to the global model for aggregation. The adjustment process aims to reduce the communication overhead and improve the convergence speed of the FL. The objective of FedHC is to minimize a loss function that depends on both the local model parameters and the local drift variables. To optimize this objective, we adopt the local drift and global model parameters updating rules of FedDC Gao et al. (2022).

### 3.4 CONVERGENCE ANALYSIS OF THE PROPOSED FEDHC

Combining cosine correlation and Hessian diagonal approximation leads to a better convergence of the proposed FedHC. The convergence of the FedHC follows from the fact that the Hessian of the objective function is positive and lipschtiz continuous, *i.e.* $H_{ii} > 0$, for all $i \in [N]$.

**Definition 3.4** *Let $\{x_n\}$ be an iterative scheme of a numerical method. Suppose $x^*$ is the original solution of the method. A number $p \geq 1$ is said to be an order of convergence of the method if there exists a $C \geq 0$ such that*

$$\lim_{n \to \infty} \frac{\|x_{n+1} - x^*\|}{\|x_n - x^*\|^p} = C$$

The following theorem ensures the better convergence of the proposed method.

**Theorem 3.2** *The order of the convergence of the FedHC is $2$.*

Additional insights regarding the thorough convergence analysis and order of convergence for the FedHC is available in the supplementary materials.

## 4 EXPERIMENTAL RESULTS AND ANALYSIS

In this section, we assess the performance of the proposed FedHC and compare it with several SOTA procedures FedDC Gao et al. (2022), FedDyn Acar et al. (2021), Scaffold Karimireddy et al. (2020), FedProx Li et al. (2020b), FedAvg McMahan et al. (2017b). We present compelling evidence of FedHC's efficacy, establishing its superiority over existing FL approaches in terms of convergence speed and model accuracy.

### 4.1 DATASETS

To test the performance of the proposed FedHC, we used the four benchmark datasets: CIFAR10, CIFAR100 Krizhevsky et al. (2009), MNIST LeCun et al. (1998), and EMNIST-L Cohen et al. (2017), over both IID and Non-IID settings. We follow the literature Gao et al. (2022) for experimental settings such as train/test split, IID, and Non-IID data division strategies. For Non-IID, we used Dirichlet data distribution with Dirichlet coefficients 0.6 and 0.3. To evaluate the proposed FedHC, we used different baseline deep learning architectures for different datasets. Particularly, we utilized fully connected CNN architectures from McMahan et al. (2017b) for MNIST, EMNIST-L, CIFAR10, and CIFAR100.

### 4.2 HYPER-PARAMETER SETTINGS

We adhered to the FL framework, wherein multiple clients independently train a global model using their respective local datasets during each communication round. Subsequently, a central server aggregates these client updates to update the global model parameters. However, for weight updation and optimization, we utilized the second-order Hessian optimizer. To preserve consistency across all techniques on the real-world datasets, we have set batch size equal to 50 for all the clients, number of local epochs of 5 for training in each communication round. We set the initial learning rate to 0.1, and the weight decay rate is 0.998. Based on the extensive experiments hyper-parameter values $\beta_1 = 0.5$, $\beta_2 = 0.9$, and $p = 0.5$ for MNIST, EMNIST-L, CIFAR10 and $\beta_1 = 0.8$, $\beta_2 = 0.9$, and $p = 0.5$ used in Hessian optimizer. $\beta_1$ delineates the first-order momentum and $\beta_2$ is second-order momentum enumerating local gradient directions, and $p$ defines the Hession power. The parameters listed above all, except weight updation adhere to the earlier works Gao et al. (2022), Acar et al. (2021), Karimireddy et al. (2020), Li et al. (2020b). We set the hyper-parameter ($\alpha$) of FedHC and FedDC as 0.01 for CIFAR10, CIFAR100, and 0.1 for MNIST. We preserve the same values for the individual hyper-parameters of the baselines as their cited studies. We adopted the Acar et al. (2021) and Li et al. (2020b) hyper-parameters value $\alpha = 0.01$ and $\mu = 10^{-4}$, respectively.

Table 1: Communication rounds required to achieve target accuracy for existing and proposed FedHC, FL approaches.

| Model | Full Participation | | | | | | Partial Participation | | | | | |
|---|---|---|---|---|---|---|---|---|---|---|---|---|
| | D1 | | D2 | | IID | | D1 | | D2 | | IID | |
| | R# | S | R# | S | R# | S | R# | S | R# | S | R# | S |
| *MNIST, 100 clients, Target Accuracy 98%* | | | | | | | | | | | | |
| FedAvg | 258 | - | 492 | - | 142 | - | 361 | - | >600 | - | 158 | - |
| FedProx | 263 | .98× | 480 | 1.03× | 136 | 1.04× | 383 | .94× | 418 | 1.44× | 149 | 1.06× |
| Scaffold | 58 | 4.45× | 58 | 8.48× | 53 | 2.68× | 62 | 5.82× | 72 | 8.33× | 50 | 3.16× |
| FedDyn | 46 | 5.61× | 51 | 9.65× | 27 | 5.26× | 122 | 2.96× | 153 | 3.92× | 71 | 2.23× |
| FedDC | 35 | 7.37× | 37 | 13.30× | 26 | 5.46× | 60 | 6.02× | 62 | 9.68× | 46 | 3.43× |
| FedSim* | 42 | 6.14× | 53 | 9.28× | 58 | 2.45× | 59 | 6.12× | 80 | 7.5× | 52 | 3.04× |
| FedAdha* | NC | NC | NC | NC | NC | NC | NC | NC | NC | NC | NC | NC |
| **Proposed** | **34** | **7.59×** | **37** | **13.30×** | **25** | **5.68×** | **42** | **8.60×** | **61** | **9.83×** | **36** | **4.39×** |
| *EMNIST-L, 100 clients, Target Accuracy 94%* | | | | | | | | | | | | |
| FedAVG | 142 | - | 192 | - | 107 | - | 153 | - | 245 | - | 108 | - |
| FedProx | 135 | 1.05× | 198 | 0.97× | 92 | 1.16× | 145 | 1.06× | 240 | 1.02× | 105 | 1.03× |
| Scaffold | 43 | 3.30× | 52 | 3.69× | 27 | 3.96× | 73 | 2.10× | 81 | 3.02× | 61 | 1.77× |
| FedDyn | **30** | **4.73×** | 52 | 3.69× | 27 | 3.96× | 73 | 2.11× | 81 | 3.02× | 61 | 1.77× |
| FedDC | 43 | 3.30× | 60 | 3.2× | 21 | 5.09× | 48 | 3.19× | 74 | 3.31× | 47 | 2.30× |
| FedSim* | 48 | 2.95× | 66 | 2.90× | 23 | 4.65× | 75 | 2.04× | 89 | 2.75× | 49 | 2.20× |
| FedAdha* | NC | NC | NC | NC | NC | NC | NC | NC | NC | NC | NC | NC |
| **Proposed** | 35 | 4.06× | **48** | **4×** | 21 | **5.10×** | **42** | **3.64×** | **55** | **4.45×** | **34** | **3.18×** |
| *CIFAR 10, 100 clients, Target accuracy 80%* | | | | | | | | | | | | |
| FedAvg | >1000 | - | >1000 | - | 286 | - | 616 | - | >1000 | - | >1000 | - |
| FedProx | 474 | 2.11× | >1000 | 1× | 277 | 1.03× | 459 | 1.34× | >1000 | 1× | 307 | 3.26 |
| Scaffold | 165 | 6.06× | 218 | 4.58× | 120 | 2.38× | 200 | 3.08× | 263 | 3.80× | 126 | 7.94× |
| FedDyn | 60 | 16.67× | 75 | 13.33× | 55 | 5.2× | 193 | 3.19× | 195 | 5.12× | 145 | 6.90× |
| FedDC | 53 | 18.86× | 70 | 14.29× | 43 | 6.65× | 141 | 4.37× | **143** | **6.99×** | 108 | 9.25× |
| FedSim* | 58 | 17.24× | 73 | 13.70× | 57 | 5.01× | 170 | 3.62× | 191 | 5.24× | 132 | 7.58× |
| FedAdha* | NC | NC | NC | NC | NC | NC | NC | NC | NC | NC | NC | NC |
| **Proposed** | **47** | **21.28×** | **59** | **16.95×** | **42** | **6.81×** | **122** | **5.05×** | 165 | 6.06× | **104** | **9.61×** |
| *CIFAR 100, 100 clients, Target accuracy 40%* | | | | | | | | | | | | |
| FedAvg | 476 | - | 847 | - | 1000 | - | 615 | - | 520 | - | 724 | - |
| FedProx | 502 | 0.93× | 507 | 1.67× | 273 | 3.66× | 280 | 2.19× | 503 | 1.03× | 650 | 1.11× |
| Scaffold | 91 | 5.31× | 94 | 9.01× | 84 | 11.90× | 106 | 5.80× | 114 | 4.56× | 113 | 6.40× |
| FedDyn | 51 | 9.16× | 53 | 15.98× | 56 | 17.86× | 149 | 4.13× | 148 | 3.51× | 143 | 5.06× |
| FedDC | 39 | 12.20× | 41 | 20.66× | **37** | **27.03×** | **102** | **6.03×** | 103 | 5.05× | **100** | **7.24×** |
| FedSim* | 54 | 8.81× | 54 | 15.68× | 58 | 17.24× | 144 | 4.27× | 160 | 3.25× | 144 | 5.03× |
| FedAdha* | NC | NC | NC | NC | NC | NC | NC | NC | NC | NC | NC | NC |
| **Proposed** | **36** | **13.22×** | **39** | **21.72×** | 39 | 25.64× | 105 | 5.86× | **103** | **5.05×** | 104 | 6.96× |

Here, 'R#', S, and NC represent the communication round, corresponding convergence speedup relative to Fed-Avg, and non-convergence, respectively. > represents the greater than operation. Whereas, D1 and D2 imply for 0.6-Dirichlet and 0.3-Dirichlet in Non-IID, settings. ∗ represents the ablation study experiments.

## 4.3 RESULTS AND ANALYSIS

We conducted extensive experiments to assess the proposed FedHC convergence speed and model performance superiority. Additionally, we demonstrated the robustness and effectiveness of the proposed FedHC across various participation levels and data heterogeneity scenarios. All our findings are presented in the context of the global model. Since the baselines and the proposed FedHC utilize the same computational resources in each round, we report the number of communication rounds rather than FLOPS. The primary objectives of FedHC are (1) accelerating the model's convergence rate to reduce communication costs and (2) enhancing the performance of models trained on diverse datasets. Our results underscore the advantages of the proposed FedHC over existing federated learning optimization approaches. Table 1 compares the convergence speed between the proposed FedHC and the aforementioned baseline methods. The results demonstrate that the proposed FedHC outperforms the other methods in effectively managing local drift and expediting convergence. The proposed FedHC achieves the target accuracy with significantly fewer communication rounds compared to McMahan et al. (2017b); Li et al. (2020b); Karimireddy et al. (2020); Acar et al. (2021); Gao et al. (2022). Moreover, the proposed FedHC achieves a reduction in the number of communication rounds by 319, 539, and 122 times with 8.56 times, 9.83 times, and 4.39 times higher speed, respectively, under D1, D2, and IID settings to attain 98% accuracy with 100 clients over partial participating settings for the MNIST dataset as compared to FedAvg. In the case of the EMNIST dataset, the proposed FedHC also achieves a reduction in the number of communication rounds by 100, 185, and 68 times with 3.4 times, 4.45 times, and 2.91 times higher speeds, respectively, un-

der D1, D2, and IID settings, resulting in 94% accuracy with 100 clients over partial participating settings as compared to FedAvg. Additionally, the proposed FedHC reduces the number of communication rounds by 494, 885, and 896 with 5.05 times, 6.06 times, and 9.61 times higher speeds, respectively, under D1, D2, and IID settings, achieving 80% accuracy with 100 clients over partial participating settings for the CIFAR-10 dataset as compared to FedAvg. Similarly, for full participation, the proposed FedHC consistently achieves equivalent or slightly better performance than FedDC regarding the number of rounds and speed across all four datasets, encompassing all three settings of D1, D2, and IID.

### 4.4 ABLATION STUDY

To examine the efficacy of the proposed FedHC, we analyze the role of the proximal gradient correction and Hessian optimizer approximation in the ablation study.

**Impact of cosine correlation:** To test the effectiveness of the cosine correlation, we conducted the experiment using an SGD optimizer with a proximal gradient correction factor FedSim. This ablation experimental results are tabulated in Table 1. From the results, it is clear that the proposed FedHC outperforms the ablation approach FedSim. To provide more specific details, FedHC substantially reduces the number of communication rounds across various scenarios. For instance, under the D1, D2, and IID settings, it accomplishes reductions of 17, 19, and 10, respectively, in conjunction with 28.81%, 23.75%, and 21.73% headway from FedSim. These improvements led to an impressive 98% accuracy with 100 clients when considering partial participation for the MNIST dataset. A similar positive impact is observed with the EMNIST dataset, where the proposed FedHC achieves reductions by 30, 34, and 12 communication rounds while maintaining 40%, 38.2%, and 24.48% higher convergence than FedSim under the respective settings. This yields a commendable 94% accuracy in the context of partial participation. Moreover, when dealing with the CIFAR-10 dataset under D1, D2, and IID settings, the proposed FedHC once again showcases its capabilities, substantially reduced by 35, 26, and 28 in communication rounds alongside 20.59%, 13.61%, and 21.21% progress. The outcome is an 80% accuracy rate with 100 clients in partial participation scenarios. For the CIFAR-100 dataset under D1, D2, and IID settings, the proposed FedHC achieved substantial reductions by 39, 57, and 40 in communication rounds alongside 27.08%, 35.62%, and 27.78% higher speeds. The outcome is a 40% accuracy rate with 100 clients' partial participation scenarios.

**Impact of approximated Hessian optimizer:** To examine the role of the approximated Hessian optimizer, we evaluated results for FedDC Gao et al. (2022) with the approximated Hessian optimizer called FedAdha. Experimental results are reported in from the results presented in Table 1, it is evident that FedAdha is not conducive to achieving convergence in FL models. Typically, first-order derivatives are commonly utilized for optimizing weight parameters. However, existing literature 202 (2020) has demonstrated that incorporating second-order derivatives can significantly improve convergence. In the ablation of FedAdha, we initially attempted to employ a second-order optimizer in conjunction with FedDC Gao et al. (2022). Regrettably, this FL approach failed to converge due to conflicts arising from the interaction between linear and non-linear optimization methods. This failure served as the impetus for us to seek an effective solution. Consequently, we introduced a novel approach: combining second-order cosine correlation with second-order approximated Hessian optimizer. This unique combination, implemented in the proposed FedHC, has proven highly effective, enabling the model to achieve superior performance while maintaining rapid convergence.

### 5 CONCLUSION

We proposed a novel proximal correction with an approximated Hessian optimizer and proximal cosine correlation for federated learning. The proposed approx. Hessian optimizer elevates and optimizes the training models with fast convergence. Cosine correlation is introduced in the objective function to mitigate the disparities between the global and client models. We have proven that integrating the Hessian diagonal operator and cosine correlation strengthens the connection between global and local models and promotes collaborative learning for fast convergence. The proposed FedHC method outperforms SOTA FL approaches on four datasets: MNIST, EMNIST, CIFAR-10, and CIFAR-100 by accelerating convergence and exhibiting superior performance across various image classification tasks, robust in both IID and Non-IID client settings.

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
