# FedHC: Proximal Correction with Hessian and Cosine Correlation for Federated Learning Supplementary Document

# Paper Id: 3657

## 1 Introduction

In this supplementary section, we first quote the notations used throughout this manuscript. Next, we will conduct a brief analysis of the objective functions in FedDC and FedDyn utilizing the Hessian optimizer. Subsequently, we present a Convergence Analysis of FedHC with a detailed discussion of the objective function's smoothness, the decreasing properties of the Hessian, and an analysis of the order of convergence. Additionally, we present ablation experimental results and analyze various settings within the FedHC across different datasets.

### 1.1 Notations

Table 1 represents all the notations utilized throughout this main manuscript and supplementary material.

Table 1: Notations and its description

| Notation | Description |
|---|---|
| $P_i$ | Restricted penalized term |
| $i$ | Represents the $i^{th}$ client |
| $d$ | Drift |
| $w$ | Global model parameters |
| $\theta$ | Local model parameters |
| $D_i$ | A Dataset on client |
| $g_i$ | Local updated value of clients' local parameters in last round |
| $g$ | Estimate update value of all client's local parameter in previous round |
| $\eta$ | Learning rate |
| $K$ | Training iterations in one round |
| $k$ | Index of training iteration in one round |
| $\gamma$ | Gradients of the loss function |
| $H$ | Hessian matrix |
| $p$ | Hessian power |
| $Diag(H)$ | Hessian diagonal |
| $L$ | Objective function |
| $\mathcal{L}$ | Empirical loss. |

### 1.2 Objective Function Analysis with Hessian in FedDC and FedDyn

The objective functions of both the FedDC and FedDyn methods incorporate various terms. In the FedDC method, the objective function encompasses an empirical loss, a restricted penalized term, and a gradient correction term as detailed in Gao et al. (2022). Specifically, the gradient correction term takes the form of $G_i(\theta_i, g_i, g) = \frac{1}{\eta K} \langle \theta_i, g_i - g \rangle$.

In the FedDyn method, the objective function includes an empirical loss, a penalized term, and dynamic regularization as explained in Acar et al. (2021). The dynamic regularization term is explicitly represented as $\langle \nabla \mathcal{L}_i(\theta_i^{t-1}), \theta \rangle$.

Importantly, both the gradient correction term $(G_i)$ in FedDC and the dynamical regularization term in FedDyn exhibit linearity due to the nature of inner products. Consequently, when applying a second-order optimizer, such as the Hessian, to either the FedDC or FedDyn objective functions, these linear terms vanish.

This absence of contribution from these linear terms becomes a crucial factor in the model weight updating process for both FedDC and FedDyn when employing the Hessian optimizer. It ultimately leads to the non-convergence of the objective functions in these methods when utilizing the Hessian optimizer, as these terms do not play a role in the optimization process. Also, from Table 5 it is evident that, using Hessian second-order optimizer on FedDC results in the non-convergence in CIFAR10 and CIFAR100 datasets.

### 1.3 CONVERGENCE ANALYSIS OF FEDHC

For a minimizing problem, we have that the first-order partial derivatives vanish at the minimized point. Further, it satisfies the second derivative test. Hence, Theorem **3.1** *(in the main manuscript)* holds. For a regressive proof of the same, the reader can refer [Jorge & Stephen (2006)]. It is well known that the order convergence of the Hessian optimizer is 2 (see [Yao et al. (2021)]). Hence, Theorem **3.2** *(in the main manuscript)* holds.

#### 1.3.1 PROOF OF LEMMA **3.1**

The cosine correlation $S_i$ is defined as:

$$S_i(w, \theta_i, g_i, g) = \frac{1}{\eta K} \left( \left( 1 - \frac{w \cdot \theta_i}{\|w\| \cdot \|\theta_i\|} \right)^2 (g - g_i) \right).$$

Now for any $w_1, w_2$ and $i \in [N]$,

$$
\begin{aligned}
\|S_i(w_1, \theta_i, g_i, g) - S_i(w_2, \theta_i, g_i, g)\| &= \left\| \frac{1}{2^k} \left[ \left( 1 - \frac{w_1 \theta_i}{\|w_1\| \|\theta_i\|} \right)^2 - \left( 1 - \frac{w_2 \theta_i}{\|w_2\| \|\theta_i\|} \right)^2 \right] \right\| \\
&= \frac{1}{2^k} \left\| \frac{\theta}{\|\theta_i\|} \left( \frac{w_1}{\|w_1\|} - \frac{w_2}{\|w_2\|} \right) \right\|^2 \\
&= \frac{1}{2^k} \|w_1 - w_2\|^2, \quad (\text{as } \|w\| \geq 0).
\end{aligned}
$$

#### 1.3.2 SMOOTHNESS OF THE OBJECTIVE FUNCTION

The objective function of each client contains three components: the local empirical loss, penalized term, and proximal gradient correction. To prove the smoothness of the objective function, we prove each of its components. Further, Lemma **3.1** ensures the smoothness of the rearmost term. Now, we provide short proof for the inaugural terms. Now for any $w_1, w_2$ and $i \in [N]$,

$$
\begin{aligned}
\|P_i(\theta_i, d_i, w_1) - P_i(\theta_i, d_i, w_2)\| &= \|(d_i + \theta_i - w_1)^2 - (d_i + \theta_i - w_2)^2\| \\
&\leq \|d_i + \theta_i - w_1\|^2 - \|d_i + \theta_i - w_2\|^2 \\
&\leq \|w_1 - w_2\|^2.
\end{aligned}
$$

Thus, the penalized term is smooth. Further, the local empirical loss satisfies the following Lipschitz property,

$$\|\mathcal{L}(\theta_i^1) - \mathcal{L}(\theta_i^2)\| \leq \beta \|\theta_i^1 - \theta_i^2\|, \quad \text{for all } \theta_i^1, \theta_i^2 \in \mathbb{R}, \text{ and } i \in [N].$$

Further, as $w$ is a function of $(\theta_i)$, we get

$$\|\mathcal{L}(w_1) - \mathcal{L}(w_2)\| \leq \alpha \|w_1 - w_2\|, \quad \text{for all } w_1, w_2 \in \mathbb{R} \text{ for some } \alpha > 0.$$

### 1.4 DECREASING PROPERTIES OF THE HESSIAN AND ORDER OF CONVERGENCE

First, we observe that the proximal gradient correction can be represented as $S_i = \frac{1}{\eta K} \left\langle \left( 1 - \frac{w \cdot \theta_i}{\|w\| \cdot \|\theta_i\|} \right), g - g_i \right\rangle$. Here $\langle \cdot, \cdot \rangle$ denotes the inner product (or dot product). As the inner product is linear with respect to linear and the $x \to x^2$ is strongly convex, we have the proximal

gradient correction is strongly convex. By assuming the local empirical loss and penalized terms are strong and as both are smooth, the objective function (Equation (**3**), *in the main manuscript*) is strongly convex and strictly smooth on the domain $\mathbb{R}^4$. Thus there exists $0 < a_1, a_2 < \infty$ such that $a_1 I \leq \nabla^2 L(\theta_i^{t,k}) \leq a_2 I$ here $t$ is the global round and $k$ is the index for local epoch. Here $X < Y$ denotes that $Y - X$ is a positive definite matrix, for two matrices $X, Y$ of the same order. Thus, we get

$$a_1 \leq \min_{j \in [N]} D_{jj} \leq \min_{j \in [N]} D_{jj} \leq a_2 \tag{1}$$

here $D_{jj}$ is the $j^{th}$ diagonal element of $Diag(H_i^{t,k,-p})$.

Now we aim to prove the updated formation local model parameter $\theta_i^{t,k}$ (Equation (**6**), *in main manuscript*) converges with respect to $t$. In particular, we show that the proper learning rate:

$$L(\theta_i^{t+1,k}) - L(\theta_i^{t,k}) \leq -\frac{a_1^k}{2a_2^{k+1}} \|\gamma_i^{t,k}\|.$$

For this, let us define $\phi_i^k : \mathbb{R} \to \mathbb{R}$ by $\phi_i^k(\theta_i^{t,k}) = \left[ \left\langle \left( H_i^{t,k,-p} \right) \gamma_i^{t,k}, \gamma_i^{t,k} \right\rangle \right]^2$. Then, we get

$$\phi_i^k(\theta_i^{t,k}) = \left[ \left\langle \left( H_i^{t,k,-p} \right) \triangle\gamma_i^{t,k}, \triangle\gamma_i^{t,k} \right\rangle \right]^2 \geq a_1^k \|\triangle\gamma_i^{t,k}\|^2. \tag{2}$$

As $L(\theta_i)$ is strongly convex, we have

$$
\begin{aligned}
L(\theta_i^{t,k} - \eta\triangle\theta_i^{t,k}) &\leq& L(\theta_i^{t,k}) - \eta\langle\triangle\theta_i^{t,k}, \gamma_i^{t,k}\rangle + \frac{\eta^2 a_1 \|\triangle\theta_i^{t,k}\|^2}{2} \\
&=& L(\theta_i^{t,k}) - \eta\phi_i^k(\theta_i^{t,k})^2 + \frac{a_2}{2a_1^k}\eta^2\phi_i^k(\theta_i^{t,k})^2, \quad \text{by Equation 2.}
\end{aligned}
$$

Therefore, by choosing an appropriate step size $\widehat{\eta} = \dfrac{a_1^k}{a_2}$, we get

$$L(\theta_i^{t,k} - \eta\triangle\theta_i^{t,k}) \leq L(\theta_i^{t,k}) - \frac{1}{2}\widehat{\eta}\phi_i^t(\theta_i^{t,k})^2.$$

Now, by using the estimate of $|Diag(H_i^{t,k,-p})|$ Equation (1) we have,

$$\phi_i(\alpha_i^{t,k}) \geq \frac{1}{a_2^k} \|\gamma_i^{t,k}\|^2.$$

Thus, we have

$$L(\theta_i^{t,k} - \eta\triangle\theta_i^{t,k}) - L(\theta_i^{t,k}) \leq -\frac{a_1^k}{2a_2^{k+1}} \|\gamma_i^{t,k}\|^2.$$

This provides the descending property of the Hessian optimizer, given in (Equation **6**, *in the main manuscript*). Now, prove that the proposed Hessian optimizer given in (Equation **7**, *in the main manuscript*) has the same convergence rate as (Equation **6**, *in the main manuscript*). For this, we first simplify our nations by $D := Diag(H_i^{t,k,-p})$. In this case, we have $j^{th}$ diagonal entry $D_{jj}$ of $D$ satisfies $\langle H_i^{t,k,-p} e_j, e_j \rangle = \langle D e_j, e_j \rangle = D_{jj}$. Here $e_j$ is the $j^{th}$ unit vector in the ordered basis (those all coordinates are zero, expert the $j^{th}$ equal to 1) for all $j \in [N]$. Then, one can easily see that the diagonal matrix $D$ as a vector and by strictly convexity of the objective function (Equation (1), we get

$$a_1 \leq D_{jj} \leq a_2, \quad \text{for all } j \in [N].$$

Therefore, each diagonal entry of $D$ in the interval $[a_1, a_2]$. Now, let us define $\psi_i^k : \mathbb{R} \to \mathbb{R}$ by $\psi_i^k(\theta_i^{t,k}) = \left[ \left\langle \left( Diag(H_i^{t,k,-p}) \right) \gamma_i^{t,k}, \gamma_i^{t,k} \right\rangle \right]^2$. Then, we get

$$\psi_i^k(\theta_i^{t,k}) = \left[ \left\langle \left( Diag(H_i^{t,k,-p}) \right) \triangle\gamma_i^{t,k}, \triangle\gamma_i^{t,k} \right\rangle \right]^2 \geq a_1^k \|\triangle\gamma_i^{t,k}\|^2. \tag{3}$$

As $L(\theta_i)$ is strongly convex, we have

$$
\begin{aligned}
L(\theta_i^{t,k} - \eta\triangle\theta_i^{t,k}) &\leq L(\theta_i^{t,k}) - \eta\langle\triangle\theta_i^{t,k}, \gamma_i^{t,k}\rangle + \frac{\eta^2 a_1\|\triangle\theta_i^{t,k}\|^2}{2} \\
&= L(\theta_i^{t,k}) - \eta\phi_i^k(\theta_i^{t,k})^2 + \frac{a_2}{2a_1^k}\eta^2\phi_i^k(\theta_i^{t,k})^2, \quad \text{by Equation 3.}
\end{aligned}
$$

Therefore, by choosing an appropriate step size $\widehat{\eta} = \dfrac{a_1^k}{a_2}$, we get

$$
L(\theta_i^{t,k} - \eta\triangle\theta_i^{t,k}) \leq L(\theta_i^{t,k}) - \frac{1}{2}\widehat{\eta}\phi_i^t(\theta_i^{t,k})^2.
$$

Now, by using the estimate of $|Diag(H_i^{t,k,-p})|$ (Equation 3) we have

$$
\psi_i(\alpha_i^{t,k}) \geq \frac{1}{a_2^k}\|\gamma_i^{t,k}\|^2.
$$

Thus, we have

$$
L(\theta_i^{t,k} - \eta\triangle\theta_i^{t,k}) - L(\theta_i^{t,k}) \leq -\frac{a_1^k}{2a_2^{k+1}}\|\gamma_i^{t,k}\|^2.
$$

This provides the descending property of the Hessian optimizer (Equation (**7**), *in the main manuscript*) and it has the same convergence rate as (Equation (**6**), *in the main manuscript*).

Further, by the strong convexity and strictly smooth properties of the objective function $L$ and employing similar techniques, we can prove the spatial averaging of the diagonal has a similar rate in the decreasing property and has the same order of convergence of (Equation (**6**), *in the main manuscript*).

## 1.5 ABLATION STUDY

To assess the efficacy of the proposed FedHC approach, we conducted an additional set of eight ablation experiments employing both full and partial (15%) client participation. These experiments encompassed three distinct settings: Non-IID (D1- 0.6 Dirichlet), Non-IID (D2- 0.3 Dirichlet), and IID, conducted across four diverse datasets: CIFAR-10, CIFAR-100, MNIST, and EMNIST-L. These extensive experiments enable us to evaluate the robustness and effectiveness of FedHC across a range of configurations and datasets, providing valuable insights into its adaptability and efficiency.

The evaluated experimental results are summarized in Tables 2 through 9. Specifically, Table 2 reports the results for CIFAR-10, presenting the number of rounds and the speed required to attain

Table 2: Communication rounds required to achieve target accuracy for existing and proposed FedHC, FL approaches on CIFAR10 with 100 clients for full participation. *The 'SpeedUp' column indicates the communication savings relative to FedAvg.*

| Model | Accuracy | Non-IID (0.6- Dirichlet) | | Non-IID (0.3 - Dirichlet) | | IID | |
|---|---|---|---|---|---|---|---|
| | | Round | SpeedUP | Round | SpeedUP | Round | SpeedUP |
| FedAvg | 0.78 | 205 | - | 346 | - | 149 | - |
| | 0.80 | >1000 | - | 1000 | - | 286 | - |
| FedProx | 0.78 | 195 | 1.05× | 350 | 0.99× | 142 | 1.05× |
| | 0.80 | >474 | 2.11× | 1000 | 1× | 277 | 1.03× |
| Scaffold | 0.78 | 123 | 1.67× | 148 | 2.34× | 89 | 1.67× |
| | 0.80 | >165 | 6.06× | 218 | 4.56× | 120 | 2.38× |
| FedDyn | 0.78 | 44 | 4.66× | 75 | 17.54× | 55 | 5.2× |
| | 0.80 | > 60 | 16.67× | 75 | 13.33× | 55 | 5.2× |
| FedDC | 0.78 | 43 | 4.77× | 53 | 6.53× | 35 | 4.25× |
| | 0.80 | 53 | 18.86× | 70 | 14.28× | 43 | 6.65× |
| Proposed | 0.78 | **35** | **5.86×** | **46** | **7.52×** | **35** | **4.25×** |
| | 0.80 | **47** | **21.27×** | **59** | **16.95×** | **42** | **6.81×** |

Table 3: Communication rounds required to achieve target accuracy for existing and proposed FedHC, FL approaches on CIFAR10 with 100 clients for partial participation (only 15% of clients are participating in every round.). *The 'SpeedUp' column indicates the communication savings relative to FedAvg.*

| Model | Accuracy | Non-IID (0.6- Dirichlet) | | Non-IID (0.3 - Dirichlet) | | IID | |
|---|---|---|---|---|---|---|---|
| | | Round | SpeedUP | Round | SpeedUP | Round | SpeedUP |
| FedAvg | 0.78 | 259 | - | 491 | - | 177 | - |
| | 0.80 | 616 | - | 1000 | - | 1000 | - |
| FedProx | 0.78 | 228 | 1.13× | 485 | 1.1× | 153 | 1.15× |
| | 0.80 | 459 | 1.34× | 1000 | 1× | 307 | 3.28× |
| Scaffold | 0.78 | 132 | 1.96× | 169 | 2.91× | 94 | 1.88× |
| | 0.80 | 200 | 3.08× | 263 | 3.80× | 126 | 7.93× |
| FedDyn | 0.78 | 118 | 2.19× | 146 | 3.39× | 110 | 1.61× |
| | 0.80 | 193 | 3.19× | 195 | 5.12× | 145 | 6.9× |
| FedDC | 0.78 | 101 | 2.56× | **105** | **4.68×** | 88 | 2.01× |
| | 0.80 | 141 | 4.37× | **143** | **6.99×** | 108 | 9.26× |
| Proposed | 0.78 | **98** | **2.64×** | 112 | 4.38× | **80** | **2.21×** |
| | 0.80 | **122** | **5.05×** | 165 | 6.06× | **104** | **9.61×** |

Table 4: Communication rounds required to achieve target accuracy for existing and proposed FedHC, FL approaches on CIFAR100 with 100 clients for full participation. *The 'SpeedUp' column indicates the communication savings relative to FedAvg.*

| Model | Accuracy | Non-IID (0.6- Dirichlet) | | Non-IID (0.3 - Dirichlet) | | IID | |
|---|---|---|---|---|---|---|---|
| | | Round | SpeedUP | Round | SpeedUP | Round | SpeedUP |
| FedAvg | 0.35 | 142 | - | 112 | - | 201 | - |
| | 0.40 | 476 | - | 847 | - | >1000 | - |
| FedProx | 0.35 | 190 | 0.75× | 124 | 0.9× | 145 | 1.39× |
| | 0.40 | 502 | 0.95× | 507 | 1.67× | 273 | 3.66× |
| Scaffold | 0.35 | 64 | 2.22× | 67 | 1.67× | 58 | 3.47× |
| | 0.40 | 91 | 5.23× | 94 | 9.01× | 84 | 11.9× |
| FedDyn | 0.35 | 38 | 3.74× | 38 | 2.95× | 45 | 4.47× |
| | 0.40 | 51 | 9.33× | 53 | 15.98× | 56 | 17.85× |
| FedDC | 0.35 | 30 | 4.73× | 33 | 3.39× | **29** | **6.93×** |
| | 0.40 | 39 | 12.2× | 41 | 20.65× | **37** | **27.03×** |
| Proposed | 0.35 | **27** | **5.26×** | **26** | **4.69×** | 31 | 6.48× |
| | 0.40 | **36** | **13.22×** | **39** | **21.72×** | 39 | 25.64× |

Table 5: Communication rounds required to achieve target accuracy for existing and proposed FedHC, FL approaches on CIFAR100 with 100 clients for partial participation (only 15% of clients are participating in every round.). *The 'SpeedUp' column indicates the communication savings relative to FedAvg.*

| Model | Accuracy | Non-IID (0.6- Dirichlet) | | Non-IID (0.3 - Dirichlet) | | IID | |
|---|---|---|---|---|---|---|---|
| | | Round | SpeedUP | Round | SpeedUP | Round | SpeedUP |
| FedAvg | 0.35 | 170 | - | 144 | | 260 | - |
| | 0.4 | 615 | - | 520 | - | 724 | - |
| FedProx | 0.35 | 227 | 0.75× | 148 | 0.97× | 187 | 1.39× |
| | 0.4 | 980 | 0.63× | 503 | 1.03× | 650 | 1.11× |
| Scaffold | 0.35 | 68 | 2.5× | 72 | 2× | 68 | 3.82× |
| | 0.4 | 106 | 5.8× | 114 | 3.56× | 113 | 6.41× |
| FedDyn | 0.35 | 98 | 1.73× | 78 | 1.46× | 106 | 2.45× |
| | 0.4 | 149 | 4.42× | 148 | 3.51× | 143 | 5.06× |
| FedDC | 0.35 | 78 | 2.18× | 74 | 1.54× | 74 | 3.51× |
| | 0.4 | **102** | **6.03×** | 103 | 5.05× | **100** | **7.04×** |
| Proposed | 0.35 | **65** | **2.62×** | **71** | **2.03×** | **71** | **3.66×** |
| | 0.4 | 105 | 5.86× | **103** | **5.05×** | 104 | 6.96× |

target accuracies of 0.78% and 0.80%, when all clients participated in the FL process. Meanwhile,

Table 6: Communication rounds required to achieve target accuracy for existing and proposed FedHC, FL approaches on MNIST with 100 clients for full participation. *The 'SpeedUp' column indicates the communication savings relative to FedAvg.*

| Model | Accuracy | Non-IID (0.6- Dirichlet) | | Non-IID (0.3 - Dirichlet) | | IID | |
|---|---|---|---|---|---|---|---|
| | | Round | SpeedUP | Round | SpeedUP | Round | SpeedUP |
| FedAvg | 0.96 | 25 | | 28 | - | 16 | - |
| | 0.98 | 258 | - | 492 | - | 142 | - |
| FedProx | 0.96 | 24 | 1.04× | 27 | 1.04× | 16 | 1× |
| | 0.98 | 263 | 0.98× | 480 | 1.03× | 136 | 1.04× |
| Scaffold | 0.96× | 11 | 2.27× | 14 | 2× | 9 | 1.78× |
| | 0.98 | 58 | 4.45× | 58 | 8.48× | 53 | 2.68× |
| FedDyn | 0.96 | 8 | 3.13× | 9 | 3.11× | 7 | 2.29× |
| | 0.98 | 46 | 5.61× | 51 | 9.65× | 27 | 5.26× |
| FedDC | 0.96 | **8** | **3.13×** | 10 | 2.8× | 7 | 2.29× |
| | 0.98 | 34 | 7.59× | 37 | 13.3× | 26 | 5.46× |
| Proposed | 0.96 | 9 | 2.78× | **9** | **3.11×** | **6** | **2.67×** |
| | 0.98 | **34** | **7.59×** | **37** | **13.3×** | **25** | **5.68×** |

Table 7: Communication rounds required to achieve target accuracy for existing and proposed FedHC, FL approaches on MNIST with 100 clients for partial participation (only 15% of clients are participating in every round.). *The 'SpeedUp' column indicates the communication savings relative to FedAvg.*

| Model | Accuracy | Non-IID (0.6- Dirichlet) | | Non-IID (0.3 - Dirichlet) | | IID | |
|---|---|---|---|---|---|---|---|
| | | Round | SpeedUP | Round | SpeedUP | Round | SpeedUP |
| FedAvg | 0.96 | 32 | - | 35 | - | 23 | - |
| | 0.98 | 361 | | > 600 | - | 158 | - |
| FedProx | 0.96 | 31 | 1.03× | 34 | 1.03× | 23 | 1× |
| | 0.98 | 383 | 0.94× | 418 | >1.44× | 149 | 1.06× |
| Scaffold | 0.96 | 20 | 1.6× | 23 | 1.52× | 16 | 1.44× |
| | 0.98 | 62 | 5.82× | 72 | 8.33× | 50 | 3.16× |
| FedDyn | 0.96 | 21 | 1.52× | 23 | 1.52× | 18 | 1.28× |
| | 0.98 | 122 | 2.96× | 153 | 3.92× | 71 | 2.23× |
| FedDC | 0.96 | 43 | 3.3× | 60 | 3.2× | 21 | 5.1× |
| | 0.98 | 78 | 3.85× | 134 | 2.24× | 50 | 6× |
| Proposed | 0.96 | **15** | **2.13×** | **17** | **2.06×** | **11** | **2.09×** |
| | 0.98 | **42** | **8.56×** | **61** | **6.85×** | **36** | **4.39×** |

Table 8: Communication rounds required to achieve target accuracy for existing and proposed FedHC, FL approaches on EMNIST-L with 100 clients for full participation. *The 'SpeedUp' column indicates the communication savings relative to FedAvg.*

| Model | Accuracy | Non-IID (0.6- Dirichlet) | | Non-IID (0.3 - Dirichlet) | | IID | |
|---|---|---|---|---|---|---|---|
| | | Round | SpeedUP | Round | SpeedUP | Round | SpeedUP |
| FedAvg | 0.94 | 142 | - | 192 | - | 107 | - |
| | 0.95 | 300 | - | 300 | - | 300 | - |
| FedProx | 0.94 | 135 | 1.05× | 198 | 0.97× | 92 | 1.16× |
| | 0.95 | 300 | 1× | 300 | 1× | 300 | 1× |
| Scaffold | 0.94 | 43 | 3.30× | 52 | 3.69× | 30 | 3.57× |
| | 0.95 | 75 | 4× | 150 | 2× | 66 | 4.55× |
| FedDyn | 0.94 | 30 | 4.73× | 52 | 3.69× | 27 | 3.96× |
| | 0.95 | 137 | 2.19× | 160 | 1.88× | 69 | 4.35× |
| FedDC | 0.94 | 43 | 3.3× | 60 | 3.2× | 21 | 5.1× |
| | 0.95 | 78 | 3.85× | 134 | 2.24× | 50 | 6× |
| Proposed | 0.94 | **35** | **4.06×** | **48** | **4×** | **21** | **5.09×** |
| | 0.95 | **76** | **4.05×** | **123** | **2.44×** | **45** | **6.67×** |

Table 3 presents the corresponding results for CIFAR-10, where only 15% of the clients were involved in the FL.

Table 9: Communication rounds required to achieve target accuracy for existing and proposed FedHC, FL approaches on EMNIST-L with 100 clients for partial participation (only 15% of clients are participating in every round.). *The 'SpeedUp' column indicates the communication savings relative to FedAvg.*

| Model | Accuracy | Non-IID (0.6- Dirichlet) | | Non-IID (0.3 - Dirichlet) | | IID | |
|---|---|---|---|---|---|---|---|
| | | Round | SpeedUP | Round | SpeedUP | Round | SpeedUP |
| FedAvg | 0.94 | 153 | - | 245 | - | 108 | - |
| | 0.95 | 300 | - | 300 | - | 300 | - |
| FedProx | 0.94 | 145 | 1.06× | 240 | 1.02× | 105 | 1.03× |
| | 0.95 | 300 | 1× | 300 | 1× | 300 | 1× |
| Scaffold | 0.94 | 44 | 3.48× | 68 | 3.6× | 42 | 2.57× |
| | 0.95 | 95 | 4.21× | 300 | 1× | 87 | 3.45× |
| FedDyn | 0.94 | 73 | 2.1× | 81 | 3.06× | 61 | 1.61× |
| | 0.95 | 127 | 2.36× | 300 | 1× | 255 | 1.18× |
| FedDC | 0.94 | 48 | 3.19× | 74 | 3.13× | 47 | 2.3× |
| | 0.95 | 92 | 3.26× | 300 | 1× | 81 | 3.7× |
| Proposed | 0.94 | **42** | **3.64×** | **55** | **4.45×** | **34** | **3.18×** |
| | 0.95 | **86** | **3.49×** | **293** | **1.02×** | **72** | **4.17×** |

Furthermore, Table 4 provides the results for CIFAR-100, including the number of rounds and speed required to achieve target accuracies of 0.35% and 0.40% with full client participation, whereas Table 5 presents the analogous results for CIFAR-100, with 15% client participation. Moving on to the MNIST dataset, Table 6 details the experimental outcomes, specifically the number of rounds and speed required to attain target accuracies of 0.96% and 0.98% when all clients participated in the FL. In contrast, Table 7 showcases the results for MNIST, where only 15% of the clients were engaged in the FL process. Finally, Tables 8 and 9 summarize the results for the EMNIST-L dataset, reporting the number of rounds and speed necessary to achieve target accuracies of 0.94% and 0.95% under two scenarios: full client participation and 15% client participation in the FL process. The results validate that the proposed FedHC outperforms almost all SOTA FL approaches across all four datasets, under various configurations and settings.