# OpenReview forum: "FedHC: Proximal Correction with Hessian and Cosine Correlation for Federated Learning"
_ICLR.cc/2024/Conference — ICLR 2024 Conference Withdrawn Submission_

### Official Review · Reviewer_YJUU · 2023-10-31

**Soundness:** 1 poor
**Presentation:** 1 poor
**Contribution:** 1 poor
**Rating:** 1
**Confidence:** 5

**Summary:**

The authors' objective is to integrate second-order information into the FL setup with local steps. To accomplish this, they first introduce a local loss function that combines the empirical loss, cosine similarity, and a penalized term to regulate the local drift. Subsequently, a second-order method is employed as a local optimizer to minimize the proposed loss function. The authors provide both theoretical convergence estimates and experimental evidence to support their approach.

**Strengths:**

1. The authors present interesting ideas, notably their application of second-order information within the context of Federated Learning (FL) with local steps.
2. The experiments conducted by the authors with 100 workers validate the applicability of the proposed method to large-scale optimization problems.

**Weaknesses:**

1. Paper is not readable. The problem and assumptions are not stated, methodology is poorly described, some notations are used without definition (like eq. (1)), theoretical analysis is not fully provided, so the correctness of the proof is not verifiable.

2. Misleading theoretical result. Theorem 3.2 according to the proof assumes strong convexity and smoothness, which is not stated in the main paper. Additionally, achieving quadratic convergence without assumptions on the initial point of the method seems doubtful, as it surpasses the lower bound for second-order methods [1].

3. Theoretical analysis is not provided. Authors only analyze the proposed scheme for one worker without incorporating second-order information in the proof. Instead, they bound the Hessian via strong convexity and smoothness constants and proceed with a typical gradient descent analysis. The analysis is presented for "local loss" $L_i^t$,  neglecting the overall loss function  ($\mathcal{L}$) that the entire procedure should minimize.
The communication aspect is entirely overlooked. In my view, the statement
"Further, by the strong convexity and strictly smooth properties of the objective function L and employing similar techniques, we can prove the spatial averaging of the diagonal has a similar rate in the decreasing property and has the same order of convergence of (Equation (6), in the main manuscript)." is insufficient to qualify as a valid proof.

4. Mathematical inaccuracies are evident, such as the misrepresentation in Definition 3.4. When $p=1$ $C$ should be in interval $(0, 1)$. In eq. 3, the vector (as per Def. 3.2) $S_i(w_i, \theta_i, g_i, g)$  is erroneously included in the loss and regularization, both of which belong to $\mathbb{R}^1$.

5. Experimental results. There is no mention of the model used in the experiments. So it is unclear wether experiments are performed on convex or non convex loss?  Furthermore, it is unclear which GPUs/CPUs was utilized, considering the authors were able to store the parameters of 101 (100 workers + 1 server) models simultaneously. I also request  the graphs with convergence, as the table does not show convergence curve. Finally, as second order information is utilized it is interesting to see wall clock time.

[1] Agarwal, N., & Hazan, E. (2018, July). Lower bounds for higher-order convex optimization. In Conference On Learning Theory (pp. 774-792). PMLR.

**Questions:**

1. What does equation 1 mean?
2. Why $\theta$ is included in Def. 3.1?
3. Theorem 3.1 is a classical optimization result and should not be highlighted as theorem in the paper.
4. Why $w_i, ~\theta_i, ~g_i, ~ g \in \mathbb{R}^4$?
5. Notation $\nabla \theta_i^{t, k}$ is misleading as $\nabla$ is vector differential operator.
6. Algorithm 1 is missing some indexes and requires rewriting, such as adding an upper index $t, ~k$ to $\theta_i$  in lines 4, 6, 7, 16.
7. It is unclear what $S$ stands for in Table 1.

---

### Official Review · Reviewer_vZhu · 2023-11-01

**Soundness:** 1 poor
**Presentation:** 1 poor
**Contribution:** 1 poor
**Rating:** 1
**Confidence:** 5

**Summary:**

The authors propose a new federated learning optimization method FedHC that incorporates second-order information through Hutchinson’s approximation. Additionally, cosine correlation is introduced. The empirical results are presented on various datasets such as MNIST, EMNIST-L, CIFAR10, and CIFAR100. The FedHC is compared with SOTA FedDC, FedDyn, Scaffold, FedProx and FedAvg.

**Strengths:**

The experimental results seem to be promising but the presentation should be improved.

**Weaknesses:**

Despite the ideas seeming interesting and perspective, the mathematical foundation and poor mathematical presentation is the main weakness of the paper.

1) Normally, the optimization problem which the paper aims to solve is presented. In this paper, it is unclear. For example, the aim could be $\min \mathbb{E} [L(\theta)]$. It also could be an average of loss functions, it could be a max of all loss functions, and so on.

2) The notation is confusing. I would recommend changing the loss function from $L(\theta)$ to $l(\theta)$ or something else, as $L$ is normally reserved for Lipschitz constants. Also, normally iteration or round counter $t$ is connected with your parameters $\theta_i^t$, not your loss.

3) Theorem 3.1 is wrong. The counter-example is function $L(x,y) = -x^2 - y^2$ and the point $(0,0)$. It is obviously a maximum of $L(x,y)$, however, the gradient is 0 and the determinant of the Hessian is positive $(-1 \cdot -1)=1$.

4) Theorem 3.2 is also wrong/not proved. The presented method doesn’t use exact Hessian information and loses a lot of the information about the curvature. So, it is not enough to say “It is well-known … see [Yao et.al. (2021)]”, which actually doesn’t claim anything similar to it in a non-federated regime.

5) Small comment: “Hessian optimizer” sounds strange to me, like you are trying to optimize the Hessian.

6) In Definition 3.2, $S_i: \mathbb{R}^4\rightarrow\mathbb{R}$, does it means, that $w_i$ is a number, not a vector? Why $w \cdot \theta_i$ multiplied as numbers here?

7) Lemma 3.1. The authors have to clarify what they mean by “smoothness” as normally it refers to Lipshitz-continuous gradient $||\nabla f(x) - \nabla f(y)|| \leq L ||x-y||$, but in Lemma 3.1 it is something different.

8) Experiments. I would recommend including more details in the Appendix about the loss and neural networks that were used to optimize. The details about the computational setup would be beneficial to understand how the experiments were performed. The code would increase the understandability and reproducibility of the experiments.  Also, it would be nice to have the convergence graphics in addition to tables to understand the convergence properties and the method’s behavior in more detail.

**Questions:**

see Weeknesses.

---

### Official Review · Reviewer_zx4S · 2023-11-03

**Soundness:** 1 poor
**Presentation:** 2 fair
**Contribution:** 1 poor
**Rating:** 3
**Confidence:** 4

**Summary:**

This paper proposes a second-order methods for FL, called FedHC, which combining proximal correction with an approximated Hessian optimizer. The paper also provides theoretical and empirical analysis for the proposed method.

**Strengths:**

Adapting second-order methods to FL is an interesting idea. The experimental results show the effectiveness of the proposed method.

**Weaknesses:**

My main concern is about the theoretical analysis. The paper does not present the assumptions on the loss function. If the loss function is nonconvex, then Thm 3.1 and Thm 3.2 are flawed. For a nonconvex function, first-order and second-order conditions can only guarantee that the point is a local minimum rather than a global minimum. Also, second-order methods may not converge to a global minimum for nonconvex functions. Even the paper assumes the loss function L is strongly convex, I don't think quadratic convergence (in Thm 3.2) can be achieved using only Hessian diagonals (in FedHC). Overall, I think the author should clarify the assumption on the loss function and provide detailed proofs of Thm 3.1 and Thm 3.2.

**Questions:**

see weakness